# Removal of clinically relevant SARS-CoV-2 variants by an affinity resin containing *Galanthus nivalis* agglutinin

**Melanie Gooldy[1], Christelle M. Roux[1], Steven P. LaRosa** [2]*, **Nicole Spaulding[1], Charles J. Fisher, Jr.[2]**

1 CUBRC, Inc., Buffalo, NY, United States of America, 2 Aethlon Medical, Inc., San Diego, CA, United States of America

* slarosamd@aethlonmedical.com

## Abstract

The Coronavirus -19 (COVID-19) pandemic due to the SARS-CoV-2 virus has now exceeded two years in duration. The pandemic has been characterized by the development of a succession of variants containing mutations in the spike protein affecting infectiousness, virulence and efficacy of vaccines and monoclonal antibodies. Resistance to vaccination and limitations in the current treatments available require the ongoing development of therapies especially for those with severe disease. The plant lectin *Galanthus nivalis* binds to mannose structures in the viral envelope. We hypothesized that viral binding should be unaffected by spike protein mutations. Known concentrations of seven clinically relevant SARS-CoV-2 variants were spiked in medium and passed three times over columns containing 1 gm of GNA affinity resin. Percent decrease in viral titer was compared with a control sample. Viral capture efficiency was found to range from 53 to 89% for all variants. Extrapolation indicated that an adult Aethlon Hemopurifier® would have more than sufficient binding capacity for viral loads observed in adult patients with severe COVID-19 infection.

## Introduction

The SARS-CoV-2 virus (COVID-19) pandemic has now exceeded 2 years in duration with 503,131,834 confirmed global cases and 6,200,571 deaths [1]. Throughout the pandemic the virus has demonstrated an ability to mutate. The mutations have occurred in the spike region of the virus [2]. Vaccines have proven beneficial in preventing severe disease, but vaccine hesitancy and non-compliance have resulted in a substantial population still at risk. Antivirals have been approved but do not have robust evidence of effectiveness in severe disease. Monoclonal antibodies have also been found to be ineffective against the most recent Omicron variant. For all these reasons, additional therapies are still needed in patients with severe COVID-19 infection. The Aethlon Hemopurifier® device contains an affinity resin containing *Galanthus nivalis* agglutinin (GNA) that binds to alpha 1,3 mannose groups in enveloped viruses [3]. It has been previously demonstrated that the Aethlon Hemopurifier®, an extracorporeal device that contains the GNA affinity resin, binds and removes SARS-CoV-2 from

**Funding:** This study was funded by the sponsor Aethlon Medical, Inc. SPL, an employee at Aethlon

Medical, Inc, adapted the clinical study from the medical literature. Aethlon Medical, Inc. funded the experiment, which was performed at CUBRC, Inc. Aethlon employees SPL and CJF analyzed the data. SPL wrote the first draft of the manuscript with input from Aethlon employee CJF.

**Competing interests:** SPL and CJF are employees and receive salaries from Aethlon Medical, Inc. which funded this study performed at CUBRC, Inc. The contributions of these authors are correct in the authors' contributions section of the submission. Specifically, SPL adapted the clinical protocol from the medical literature. SPL also analyzed the data and wrote the first draft of the manuscript. Author CJF also analyzed the data and made comments on the manuscript draft. Aethlon Medical, Inc funded this study and is the manufacturer of the Hemopurifier, an experimental device in clinical development that contains the affinity resin tested in this study. This commercial affiliation does not alter our adherence to PLOS ONE policies on sharing data and materials.

the circulation in vivo [4]. As the mutations in the SARS-CoV-2 should not affect mannosylation in the viral envelope, we hypothesized that the GNA affinity resin should have similar binding to the major variants causing clinical disease over the past two years. We challenged small columns containing the affinity resin with known concentrations of SARs-CoV-2 variants and calculated percent binding.

## Materials and methods

### Viruses

Seven variants of SARS-CoV-2, (Table 1) were propagated on Vero E6 cells (CRL-1586) and Calu-3 cells (HTB-55™). They were all propagated in EMEM medium supplemented with 2% FBS in 5% CO2, at an MOI of 0.001 for 2 days. Cell culture supernatants were collected, clarified by centrifugation, aliquoted and stored at -80˚C until use. For the last propagation, infections were performed in EMEM supplemented with 2% exosome-free FBS. Titrations of viral stocks were performed by plaque assay.

### GNA affinity resin, column preparation and challenge

*Galanthus nivalis* agglutinin was covalently bonded to diatomaceous earth via a process previously described [5]. A total of 1 gm of affinity resin was added to each column. The experimental protocol has been adapted from a previous in vitro experiment testing the binding of pathogens including SARS-CoV-2 by media in an extracorporeal cartridge [6, 7]. For each SARS-CoV-2 variant tested, 4 columns were prepared (3 with affinity resin and 1 no resin control). The column set up is depicted in Fig 1A. Resin was first washed once with 10 ml, then with 5 ml Phosphate Buffered Saline (PBS). Each variant was diluted in EMEM and 2% exosome-free FBS to achieve viral concentrations of approximately $1x10^4$ PFU/mL, such that a 5-mL aliquot would provide a total challenge of $5.0 \times 10^4$ PFU/column. This viral challenge is similar to what was used in in vitro studies of the inactivation of SARS-CoV-2 by UV light and riboflavin in biologic fluids [8, 9]. Previous studies have estimated the ratio of PFUs/ml to viral RNA copies/ml to be about 1:3400 [10]. This would equate to a total viral challenge of approximately $1.7x 10^8$ copies ($3.4 \times 10^7$ copies/ml). The study our experimental protocol was adapted from utilized a viral challenge of $9.35 X 10^8$ copies/ml [7].

5 mL of viral suspension was transferred dropwise to the top and side of the column containing the affinity resin bed (Fig 1B). Once the challenge suspension passed through the column, the effluent was passed through two additional times. Simultaneously, the control column received three sequential 5-mL passages of challenge suspension using the same procedure. The viral titer of each challenge suspension and the collected samples after 3 passages were analyzed for presence of viable virus using plaque assay.

### Sample analysis by plaque assay

Samples were serially diluted, and triplicate aliquots of each dilution were transferred onto confluent monolayers of VeroE6 cells (12-well plate format). The plates were incubated at 37˚C with 5% $CO_2$ for one hour with $CO_2$ and gently rocked every 15 minutes to promote virus adsorption. After the initial 1-hour incubation, the dilution aliquots were removed from each well and an overlay of microcrystalline cellulose (0.4% to 0.75%) was added to each well. The plates were incubated at 37˚C for 96 to 144 hours, depending on the variant tested. After incubation, the microcrystalline cellulose overlays were removed, cells were fixed, and viruses inactivated with 10% formalin for 1 hour. The formalin was then removed, wells were washed with water, stained with crystal violet (15 minutes). After removal of crystal violet, each well was washed with water, the plates were allowed to dry, and the plaques were counted in each well.

**Table 1. SARS-CoV-2 variants.**

| BEI No. | Variant Description* |
|---------|----------------------|
| NR-54009 | SARS-CoV-2, Isolate hCoV-19/South Africa/KRISP-K005325/2020 |
| NR-54982 | SARS-CoV-2, isolate hCoV-19/Japan/TY7-503/2021 (Lineage Brazil P.1) [#] |
| NR-54000 | SARS-CoV-2, isolate hCov-19/England/204820464/2020 (Lineage B.1.1.7) [##] |
| NR-55672 | SARS-CoV-2, Isolate hCoV-19/USA/MD-HP05647/2021 (Lineage B.1.617.2; Delta Variant)[###] |
| NR-55691 | SARS-CoV-2, Isolate hCoV-19/USA/CA-VRLC086/2021 (Lineage AY.1; Delta Variant) [###] |
| NR-55654 | SARS-CoV-2, Isolate hCoV-19/Peru/un-CDC-2-4069945/2021 (Lineage C.37; Lambda Variant) |
| NR-56461 | SARS-CoV-2, Isolate hCoV-19/USA/MD-HP20874/2021 (Lineage B.1.1.529; Omicron Variant)[###] |

*Variants were deposited by the Centers for Disease Control and Prevention and obtained through BEI Resources, NIAID, NIH

[#] contributed by National Institute of Health;

[##] Bassam Hallis and

[###] Dr. Andrew S. Pekosz

## Calculation of GNA affinity resin binding

The number of viable organisms in the suspension after passages over the resin bed were used to perform calculations of resin efficacy. The amount of viable virus collected was compared to that of the column control to calculate capture efficiency. Percent reductions were calculated as follows:

$$Percent\ Reduction = (1 - (B/A)) \times 100\%$$

where:

$A$ = number of viable organisms per milliliter recovered from the column control sample

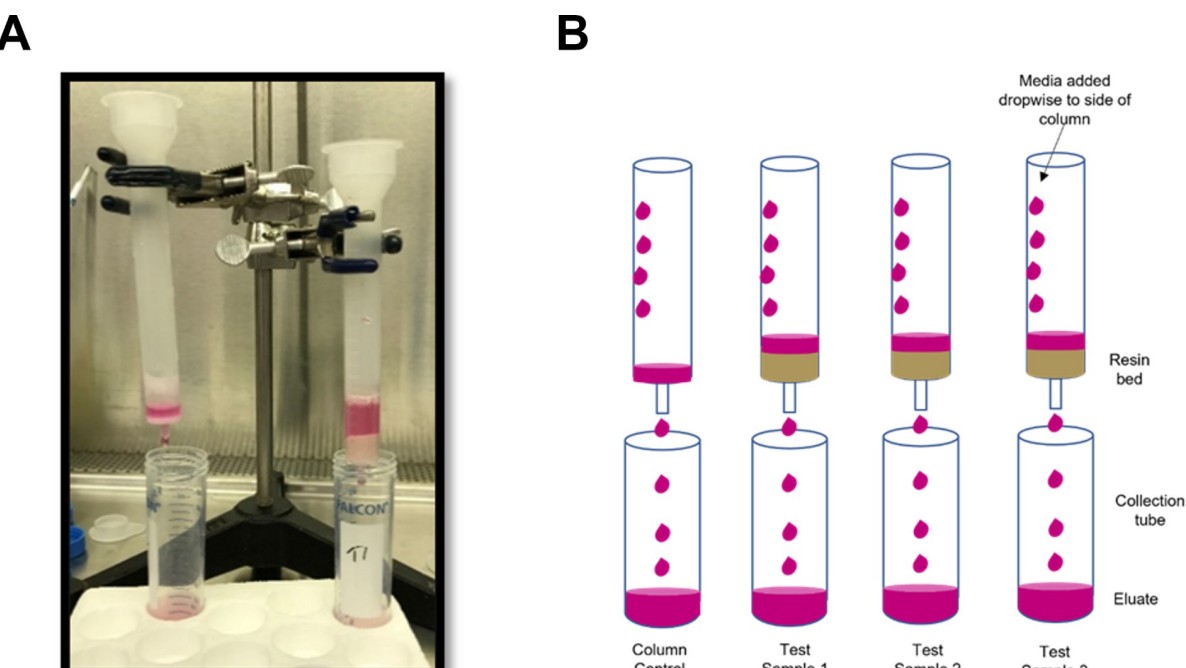

**Fig 1. Experimental set-up.** a. Set-up of columns in clamp/ ring stand. b. Schematic diagram.

*B* = number of viable organisms per milliliter recovered from the test samples

## Results

The *Galanthus nivalis* agglutinin (GNA) affinity resin demonstrated capture efficiencies ranging from 53.2% to 89.9% for the seven SARS-CoV-2 variants tested (Table 2). The resin columns were successful at removing greater than 70% of the viral load in a single pass for four of the seven variants.

The following summary Tables 3–9 present the detailed data sets for each variant. The tables present the concentration, in PFU/mL, of the column control (without resin), and each of the three test samples collected during the experiments and the calculated percent capture efficiency.

During the NR 54009 test, the 5-mL challenge aliquot for Sample 2 passed through the column in almost half the time than that of samples 1 and 3. While no air pockets in the resin bed were visible, it is possible that some channeling of the challenge suspension occurred. This may account for the lower capture efficiency observed for sample 2.

## Discussion

In this in vitro model, we demonstrated that the *Galanthus nivalis* Agglutinin affinity resin present in the Aethlon Hemopurifier device can bind all the major COVID-19 variants to date. Some variability was observed with the greatest binding observed for the Brazilian P.1 variant and current Omicron variant and the lowest observed for the Delta variant AY.1 ("Delta Plus"). This data correlates with the in vivo data with the adult Hemopurifier where a COVID-19 patient experienced a 58% reduction in plasma viral load following a single 6-hour treatment [4].

The total viral challenge to each column in our study was approximately $1.7 \times 10^8$ copies. Taking the most conservative data point of 53.2% viral removal this would equate to a viral removal of 90,440,000 copies for 1 gram of GNA affinity resin. An adult Hemopurifier contains 40 grams of affinity resin giving a total binding capacity of $3.62 \times 10^9$ of viral copies. The average blood RNA viral load for a severely ill COVID patient is usually $10^3$ copies/ml. Assuming a viral load of 5000 copies/ ml in an adult patient with a blood volume of 5000 ml would equate to a total viral load of $2.5 \times 10^7$ copies indicating that the Hemopurifier would have more than sufficient binding capacity.

A number of recently published studies have highlighted the importance of SARS-CoV-2 viremia to the pathogenesis of the disease. In a study by Bermejo-Martin and colleagues, the presence of viremia was associated with a dysregulated immune response and development of coagulopathy [11]. In a meta-analysis of 21 studies involving 2181 patients, SARS-CoV-2 RNAemia was associated with intensive stay and poor outcome [12]. A study from Wuhan

**Table 2. Average column capture efficiency for SARS-CoV-2 variants.**

| Variant ID | Capture efficiency (%) |
|---|---|
| NR 54009 (South Africa) | 69.3 ± 11.4 |
| NR 54000 (UK) | 69.8 ± 4.7 |
| NR 54982 (Brazil) | 89.0 ± 3.7 |
| NR 55672 (B.1.672 Delta) | 78.8 ± 1.9 |
| NR 55654 (Lambda) | 70.5 ± 3.6 |
| NR 55691 (AY.1 Delta) | 53.2 ± 11.6 |
| NR 56461 (Omicron) | 89.9 ± 2.1 |

**Table 3. NR-54009: Isolate hCoV-19/South Africa/KRISP-K005325/2020.**

| Sample description | Passage time (sec) | Concentration (PFU/mL) | Reduction (%) |
|---|---|---|---|
| Column control | 90 | $2.97 \times 10^4$ | - |
| Test sample 1 | 85 | $7.00 \times 10^3$ | 76.4 |
| Test sample 2 | 40 | $1.30 \times 10^4$ | 56.2 |
| Test sample 3 | 90 | $7.33 \times 10^3$ | 75.3 |
| Average: | $71.7 \pm 27.5$ | | $69.3 \pm 11.4$ |

**Table 4. NR-54000: Isolate hCov-19/England/204820464/2020 (Lineage B.1.1.7).**

| Sample description | Passage time (sec) | Concentration (PFU/mL) | Reduction (%) |
|---|---|---|---|
| Column control | 55 | $1.27 \times 10^4$ | - |
| Test sample 1 | 80 | $3.87 \times 10^3$ | 69.5 |
| Test sample 2 | 82 | $4.40 \times 10^3$ | 65.3 |
| Test sample 3 | 83 | $3.20 \times 10^3$ | 74.7 |
| Average: | $81.7 \pm 1.5$ | | $69.8 \pm 4.7$ |

**Table 5. NR-54982: Isolate hCoV-19/Japan/TY7-503/2021 (Lineage Brazil P.1).**

| Sample description | Passage time (sec) | Concentration (PFU/mL) | Reduction (%) |
|---|---|---|---|
| Column control | 55 | $4.93 \times 10^3$ | - |
| Test sample 1 | 90 | $6.20 \times 10^2$ | 87.4 |
| Test sample 2 | 93 | $6.73 \times 10^2$ | 86.4 |
| Test sample 3 | 105 | $3.33 \times 10^2$ | 93.2 |
| Average: | $96.0 \pm 7.9$ | | $89.0 \pm 3.7$ |

**Table 6. NR-55672: Isolate hCoV-19USA/MD-HP05647/2021 (Lineage B.1.617.2; Delta Variant).**

| Sample description | Passage time (s) | Concentration [PFU/mL] | Reduction (%) |
|---|---|---|---|
| Column control | 45 | $2.47 \times 10^4$ | - |
| Test sample 1 | 54 | $5.73 \times 10^3$ | 76.8 |
| Test sample 2 | 71 | $5.13 \times 10^3$ | 79.2 |
| Test sample 3 | 108 | $4.80 \times 10^3$ | 80.5 |
| Average: | $77.7 \pm 27.6$ | | $78.8 \pm 1.9$ |

**Table 7. NR-55654: Isolate hCoV-19/Peru/un-CDC-2-4069945/2021 (Lineage C.37; Lambda Variant).**

| Sample description | Passage time (sec) | Concentration (PFU/mL) | Reduction (%) |
|---|---|---|---|
| Column control | 42 | $1.67 \times 10^3$ | - |
| Test sample 1 | 61 | $4.87 \times 10^2$ | 70.8 |
| Test sample 2 | 70 | $4.33 \times 10^2$ | 74.0 |
| Test sample 3 | 72 | $5.53 \times 10^2$ | 66.8 |
| Average: | $67.7 \pm 5.9$ | | $70.5 \pm 3.6$ |

**Table 8. NR-55691: Isolate hCoV-19/USA/CA-VRLC086/2021 (Lineage AY.1; Delta Variant).**

| Sample description | Passage time (sec) | Concentration (PFU/mL) | Reduction (%) |
|---|---|---|---|
| Column control | 24 | $5.27 \times 10^3$ | - |
| Test sample 1 | 65 | $2.33 \times 10^3$ | 55.7 |
| Test sample 2 | 55 | $3.13 \times 10^3$ | 40.5 |
| Test sample 3 | 61 | $1.93 \times 10^3$ | 63.3 |
| Average: | $60.3 \pm 5$ | | $53.2 \pm 11.6$ |

China found that organ failure damage (respiratory failure, cardiac damage, renal damage, and coagulopathy) was more common in patients with SARS-CoV-2 RNAemia than those without [13]. Finally, two studies have implicated the presence of SARS-CoV-2 viremia with the development of Post-Acute Sequelae of COVID-19 (PASC) [14, 15] These studies raise the hypothesis that removal of SARS-CoV-2 from the bloodstream could potentially improve clinical outcomes in severe disease.

An extracorporeal therapy for COVID-19 viremia may help overcome limitations in the current treatment strategies. Vaccines have been remarkably effective in decoupling infection from severe disease. However, recent data indicates that only 65% of the population have been vaccinated and only 30% have received booster shots [16]. Vaccines have also demonstrated decreased neutralization against the current Omicron variant [17]. Three antiviral drugs are currently approved for COVID-19 with only Remdesivir having been studied in severe disease. Remdesivir showed improvement in time to recovery but only in patients not on mechanical ventilation [18]. The Infectious Disease Society of America (IDSA) does not currently recommend Remdesivir for COVID-19 patients on mechanical ventilation based on the available data [19]. Monoclonal therapies have been developed during the pandemic but recently combination therapies of bamlanivimab/etsevemivab and casirivimab/indemivab have been found to have reduced activity against the Omicron BA.2 variant [20]. Sotrovimab is the only monoclonal antibody currently recommended in the NIH guidelines [21]. Anti-inflammatory therapies used in COVID-19 include the corticosteroid Dexamethasone and IL-6 inhibitor Tocilizumab. Corticosteroid exposure is associated with gastrointestinal bleeding and can lead to opportunistic infections. Tocilizumab has only a single target and previously been associated with gastrointestinal perforations and increased risk of infections [22, 23]. The GNA affinity resin in the Hemopurifier may provide an alternative approach to modulating the immune and coagulopathic response to COVID-19 through removal of exosomes with microRNAs associated with acute lung injury and coagulopathy [3].

The study presented has limitations. The in vitro system set up does not replicate the adult Hemopurifier used in humans. As the columns tested were packed with 1 gram of affinity resin over which a viral challenge was passed, the flow rates, viral dwell times and physical contact with the affinity resin are not the same as would be observed in a human being treated with the adult Hemopurifier. Furthermore, the viral challenge was diluted in culture media

**Table 9. NR-56461: Isolate hCoV-19/USA/MD-HP20874/2021 (Lineage B.1.1.529; Omicron Variant).**

| Sample description | Passage time (sec) | Concentration (PFU/mL) | Reduction (%) |
|---|---|---|---|
| Column control | 27 | $6.00 \times 10^3$ | - |
| Test sample 1 | 65 | $7.07 \times 10^2$ | 882 |
| Test sample 2 | 70 | $7.67 \times 10^2$ | 87.2 |
| Test sample 3 | 63 | $5.20 \times 10^2$ | 91.3 |
| Average: | $66.0 \pm 3.6$ | | $88.9 \pm 2.1$ |

and not in a biologic fluid. Lastly, a column containing the diatomaceous earth without the GNA was not utilized as a control to the assess the contributions of the individual components of the affinity resin to viral binding. Efforts are underway to construct a mini-Hemopurifier to approximate the clinical situation. A clinical trial is currently underway in severe and critically ill COVID-19 infected patients with the Aethlon Hemopurifier containing this affinity resin.

## Conclusions

A column containing a GNA affinity resin can bind all the major COVID-19 variants causing clinical disease. This technology will likely remain active against future COVID-19 variants that affect the efficacy of vaccines and treatments.

## Author Contributions

**Conceptualization:** Steven P. LaRosa, Charles J. Fisher, Jr.

**Data curation:** Melanie Gooldy, Christelle M. Roux, Nicole Spaulding.

**Formal analysis:** Melanie Gooldy, Christelle M. Roux, Steven P. LaRosa, Nicole Spaulding.

**Investigation:** Melanie Gooldy, Christelle M. Roux.

**Methodology:** Steven P. LaRosa.

**Writing – original draft:** Steven P. LaRosa.

**Writing – review & editing:** Melanie Gooldy, Christelle M. Roux, Nicole Spaulding, Charles J. Fisher, Jr.

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
