## [Decision Letter · Decision Letter 0]

17 Jun 2022

PONE-D-22-13008Removal of Clinically Relevant SARS-CoV-2 Variants by An Affinity Resin Containing Galanthus nivalis AgglutininPLOS ONE

Dear Dr. LaRosa, 

Thank you for submitting your manuscript to PLOS ONE. After careful consideration, we feel that it has merit but does not fully meet PLOS ONE’s publication criteria as it currently stands. Therefore, we invite you to submit a revised version of the manuscript that addresses the points raised during the review process. Both reviewers believe that your manuscript is interesting. However, they have some concerns that need to be addressed. 

We look forward to receiving your revised manuscript.

Kind regards,

Gheyath K. Nasrallah

Academic Editor

PLOS ONE

Journal Requirements:

SPL and CJF are employees of Aethlon Medical, Inc.  

We note that one or more of the authors have an affiliation to the commercial funders of this research study: Aethlon Medical, Inc. 

Reviewers' comments:

Reviewer's Responses to Questions

**Comments to the Author**

1. Is the manuscript technically sound, and do the data support the conclusions?

Reviewer #1: Yes

Reviewer #2: Yes

2. Has the statistical analysis been performed appropriately and rigorously? 

Reviewer #1: I Don't Know

Reviewer #2: Yes

3. Have the authors made all data underlying the findings in their manuscript fully available?

Reviewer #1: Yes

Reviewer #2: Yes

4. Is the manuscript presented in an intelligible fashion and written in standard English?

Reviewer #1: Yes

Reviewer #2: Yes

5. Review Comments to the Author

Reviewer #1: First of all I would like to congratulate the whole research team for the creativity, authenticity and suitability of the idea. However, I would like to share with you some questions related mainly to the methodology and results:

1. Taking into account the impact that variability due to random error could have on the results and conclusions, it would be necessary to argue why you have only performed 3 experiments (excluding the control) for each strain and not a minimum previously calculated in order to try to reduce the effects of such variability.

2. I note that when comparing the problem samples with the control, the percentage of viral reduction in the columns with the resin was compared to a column that allows the free flow of SARS-CoV-2. It would be necessary for the authors to argue why they have not used as a control a resin without mannan-binding lectin but made of a material with physical-chemical characteristics similar to the resin evaluated.

3. It would be necessary to explain if there is an objective justification (reference or rationale) that has led you to set the viral concentration in the columns at 5 x 104 PFU/ml since the amount of virus per unit of volume will probably influence the percentage of viral reduction and thus the results.

4. It would be necessary to explain if there is an objective justification (reference or rationale) that has led you to pass the test sample 3 times through the resin since the number of times this phenomenon occurs will certainly affect the percentage of viral reduction and thus the results.

5. From the information in the tables could be assume that the percentage of viral reduction in the control column is 0%?

In summary, Nobody could deny that the time of the fluid flow through the column, physical-chemical characteristics of the resin, viral concentration in the column, number of times of passage and the absence of any type of resin in the control columns are critical factors in order to support the findings, so it could be good for your research to elaborate a paragraph in the discussion to record some of the possible limitations regarded below.

Reviewer #2: In this paper Gooldy et al. present a research article on reduction of seven SARS-CoV-2 variants by a Galanthus nivalis agglutinin-based affinity resin (Hemopurifier resin).

The study presents the results of interesting primary scientific research, which according to PubMed was not published elsewhere up to now. The analyses are technically/virologically sound and sufficiently described.

Besides not also testing reduction from plasma the article is scientifically well done. However this issue is acceptable because of in vivo Hemopurifier clearance data (58% reduction of plasma viral load) from a severely ill COVID-19 patient (Amundson et al. 2021, Ref. 4).

The data associating viremia to severity of COVID-19 are nicely discussed.

The work is well organized. English language and stlye are fine. I recommend to publish the article after the correction of following minor issues:

- The SARS-CoV-2 variant from the in vivo treatment report (Amundson et al.) should be mentioned (Delta variant?).

- "A clinical trial is currently underway in severe and critically ill COVID-19 infected patients with the Aethlon Hemopurifier containing this affinity resin" should be removed from the conclusion and better placed in the discussion, as the trial is first mentioned in the conclusion.

- the figures might be condensed into one. Fig. 1A: ring stand photo, 1B: column test set-up.

6. PLOS authors have the option to publish the peer review history of their article (what does this mean?). If published, this will include your full peer review and any attached files.

Reviewer #1: No

Reviewer #2: No

---

## [Author Response · Author response to Decision Letter 0]

11 Jul 2022

1 July 2022

To Editor, PLOS ONE

From: Steven P. LaRosa, MD

Re: Response to Reviewers PONE-D-22-13008

Dear Editor,

Below please see our point-by-point response to each Reviewer’s comments:

Reviewer #1: First of all I would like to congratulate the whole research team for the creativity, authenticity and suitability of the idea. However, I would like to share with you some questions related mainly to the methodology and results:

1. Taking into account the impact that variability due to random error could have on the results and conclusions, it would be necessary to argue why you have only performed 3 experiments (excluding the control) for each strain and not a minimum previously calculated in order to try to reduce the effects of such variability.

Authors’ Response: Our experimental protocol was adapted from 2 previous publications of a media in an extracorporeal cartridge examining the in vitro removal of pathogens. In these studies, three replicates were performed for each pathogen. Additionally, the pathogen challenge was passed over the mini columns three times in these studies. 

Language has been added to the methods section (lines 80-82) and the new references (6,7) have been added to the revised manuscript.

2. I note that when comparing the problem samples with the control, the percentage of viral reduction in the columns with the resin was compared to a column that allows the free flow of SARS-CoV-2. It would be necessary for the authors to argue why they have not used as a control a resin without mannan-binding lectin but made of a material with physical-chemical characteristics similar to the resin evaluated.

Authors’ Response: This study was designed to assess the binding of the affinity resin in adult Aethlon Hemopurifier. This resin contains GNA covalently bound to diatomaceous earth. We did not assess the viral binding of the diatomaceous earth without the GNA. We have included this as a limitation of the study (lines 209-211).

3. It would be necessary to explain if there is an objective justification (reference or rationale) that has led you to set the viral concentration in the columns at 5 x 104 PFU/ml since the amount of virus per unit of volume will probably influence the percentage of viral reduction and thus the results.

Authors’ Response: we wanted to set the viral challenge such that it would be higher than the typical COVID-19 viral load in patient such that we could assess the total binding capacity of resin. The viral challenge of 1 x 104 PFU/ml (3.40 x 107 copies/ml) is similar to the COVID-19 challenge that has been used in in vitro studies of SARS-CoV-2 inactivation in biologic fluids by riboflavin and UV light (new references 8 and 9). A viral challenge of 9.35 X 108 copies/ml was used in the experiment our study was adapted from. (Olson SW, Oliver JD, Collen J, et al. Treatment of Severe Coronavirus Disease 2019 With the Seraph-100 MicroBind Affinity Blood Filter. Crit Care Expl 2020;2:e0180.)

The new language (lines 86-80) and new references (8,9) are found in the revised manuscript.

4. It would be necessary to explain if there is an objective justification (reference or rationale) that has led you to pass the test sample 3 times through the resin since the number of times this phenomenon occurs will certainly affect the percentage of viral reduction and thus the results.

Authors’ Response: Our experimental protocol was adapted from 2 previous publications of a media in an extracorporeal cartridge examining the in vitro removal of pathogens. In these studies, three replicates were performed for each pathogen. Additionally, the pathogen challenge was passed over the mini columns three times in these studies. 

Language has been added to the methods section (lines 77-79) and references (6 and 7) have been added to the revised manuscript.

5. From the information in the tables could be assume that the percentage of viral reduction in the control column is 0%?

Authors’ Response: That is correct.

In summary, Nobody could deny that the time of the fluid flow through the column, physical-chemical characteristics of the resin, viral concentration in the column, number of times of passage and the absence of any type of resin in the control columns are critical factors in order to support the findings, so it could be good for your research to elaborate a paragraph in the discussion to record some of the possible limitations regarded below.

Authors’ Response: The paragraph containing the limitations of the study (lines 196-204) has been revised to reflect the reviewer’s comments.

Reviewer #2: In this paper Gooldy et al. present a research article on reduction of seven SARS-CoV-2 variants by a Galanthus nivalis agglutinin-based affinity resin (Hemopurifier resin).

The study presents the results of interesting primary scientific research, which according to PubMed was not published elsewhere up to now. The analyses are technically/virologically sound and sufficiently described.

Besides not also testing reduction from plasma the article is scientifically well done. However, this issue is acceptable because of in vivo Hemopurifier clearance data (58% reduction of plasma viral load) from a severely ill COVID-19 patient (Amundson et al. 2021, Ref. 4).

The data associating viremia to severity of COVID-19 are nicely discussed.

The work is well organized. English language and stlye are fine. I recommend to publish the article after the correction of following minor issues:

- The SARS-CoV-2 variant from the in vivo treatment report (Amundson et al.) should be mentioned (Delta variant?).

Authors’ Response: This patient’s clinical specimen was not sent out for genetic analysis to determine which variant it was. The timing of this infection would suggest it was likely the alpha variant.

- "A clinical trial is currently underway in severe and critically ill COVID-19 infected patients with the Aethlon Hemopurifier containing this affinity resin" should be removed from the conclusion and better placed in the discussion, as the trial is first mentioned in the conclusion.

Authors’ Response: The authors agree. Please see revised manuscript (lines 202-204).

- the figures might be condensed into one. Fig. 1A: ring stand photo, 1B: column test set-up.

Author’s Response: The authors agree. Please see revised manuscript and figures have been resubmitted as new figure 1.

---

## [Decision Letter · Decision Letter 1]

19 Jul 2022

Removal of Clinically Relevant SARS-CoV-2 Variants by An Affinity Resin Containing Galanthus nivalis Agglutinin

PONE-D-22-13008R1

Dear Dr. LaRosa,

We’re pleased to inform you that your manuscript has been judged scientifically suitable for publication and will be formally accepted for publication once it meets all outstanding technical requirements.

Kind regards,

Hana Maria Dobrovolny, Ph.D

Academic Editor

PLOS ONE

Additional Editor Comments (optional):

Reviewers' comments:

Reviewer's Responses to Questions

**Comments to the Author**

1. If the authors have adequately addressed your comments raised in a previous round of review and you feel that this manuscript is now acceptable for publication, you may indicate that here to bypass the “Comments to the Author” section, enter your conflict of interest statement in the “Confidential to Editor” section, and submit your "Accept" recommendation.

Reviewer #1: All comments have been addressed

Reviewer #2: All comments have been addressed

2. Is the manuscript technically sound, and do the data support the conclusions?

Reviewer #1: Yes

Reviewer #2: Yes

3. Has the statistical analysis been performed appropriately and rigorously? 

Reviewer #1: Yes

Reviewer #2: Yes

4. Have the authors made all data underlying the findings in their manuscript fully available?

Reviewer #1: Yes

Reviewer #2: Yes

5. Is the manuscript presented in an intelligible fashion and written in standard English?

Reviewer #1: Yes

Reviewer #2: Yes

6. Review Comments to the Author

Reviewer #1: The authors' responses and the modifications to the text were in accordance with the requirements, so once again we can only congratulate them for their efforts.

Reviewer #2: (No Response)

7. PLOS authors have the option to publish the peer review history of their article (what does this mean?). If published, this will include your full peer review and any attached files.

Reviewer #1: No

Reviewer #2: No

---

## [Editor Report · Acceptance letter]

22 Jul 2022

PONE-D-22-13008R1 

Removal of clinically relevant SARS-CoV-2 variants by an affinity resin containing *Galanthus nivalis* agglutinin 

Dear Dr. LaRosa:

I'm pleased to inform you that your manuscript has been deemed suitable for publication in PLOS ONE. Congratulations! Your manuscript is now with our production department. 

Kind regards, 

on behalf of

Dr. Hana Maria Dobrovolny 

Academic Editor

PLOS ONE